# Oxidative Products of Curcumin Rather Than Curcumin Bind to *Helicobacter Pylori* Virulence Factor VacA and Are Required to Inhibit Its Vacuolation Activity

**DOI:** 10.3390/molecules27196727

**Published:** 2022-10-09

**Authors:** Maya Chaturvedi, Mohit Mishra, Achyut Pandey, Jyoti Gupta, Jyoti Pandey, Shilpi Gupta, Md. Zubbair Malik, Pallavi Somvanshi, Rupesh Chaturvedi

**Affiliations:** 1School of Biotechnology, Jawaharlal Nehru University, New Delhi 110067, India; 2Department of Energy and Environment, TERI School of Advanced Studies, New Delhi 110067, India; 3Department of Genetics and Bioinformatics, Dasman Diabetes Institute, Dasman, P.O. Box 1180, Kuwait City 15462, Kuwait; 4School of Computational and Integrative Sciences, Jawaharlal Nehru University, New Delhi 110067, India; 5Special Center for System Medicine, Jawaharlal Nehru University, New Delhi 110067, India; 6Nanofluidiks Pvt. Ltd., Jawaharlal Nehru University-Foundation for Innovation, New Delhi 110067, India

**Keywords:** curcumin, *H. pylori*, VacA, vacuolation, oxidation

## Abstract

Curcumin is a hydrophobic polyphenol derived from turmeric with potent anti-oxidant, anti-microbial, anti-inflammatory and anti-carcinogenic effects. Curcumin is degraded into various derivatives under in vitro and in vivo conditions, and it appears that its degradation may be responsible for the pharmacological effects of curcumin. The primary risk factor for the cause of gastric cancer is *Helicobacter pylori (H. pylori)*. A virulence factor vacuolating cytotoxic A (VacA) is secreted by *H. pylori* as a 88 kDa monomer (p88), which can be fragmented into a 33 kDa N-terminal domain (p33) and a 55 kDa C-terminal domain (p55). Recently it has been reported that curcumin oxidation is required to inhibit the activity of another major *H.pylori* toxin CagA. We performed molecular docking of curcumin and its oxidative derivatives with p33 and p55 domains of VacA. Further, we have examined the effect of the oxidation of curcumin on the vacuolation activity of VacA protein. We observed the binding of curcumin to the p55 domain of VacA at five different sites with moderate binding affinities. Curcumin did not bind to p33 domain of VacA. Remarkably, cyclobutyl cyclopentadione and dihydroxy cyclopentadione, which are oxidized products of curcumin, showed a higher binding affinity with VacA protein at all sites except one as compared to parent curcumin itself. However, cyclobutyl cyclopentadione showed a significant binding affinity for the active site 5 of the p55 protein. Active site five (312–422) of p55 domain of VacA plays a crucial role in VacA-mediated vacuole formation. Invitro experiments showed that curcumin inhibited the vacuolation activity of *H. pylori* in human gastric cell line AGS cells whereas acetyl and diacetyl curcumin, which cannot be oxidized, failed to inhibit the vacuolation in AGS cells after *H. pylori* infection. Here our data showed that oxidation is essential for the activity of curcumin in inhibiting the vacuolation activity of *H. pylori*. Synthesis of these oxidized curcumin derivatives could potentially provide new therapeutic drug molecules for inhibiting *H. pylori*-mediated pathogenesis.

## 1. Introduction

Curcumin, a natural polyphenolic compound derived from Curcuma species, has gained particular interest because of its inhibitory effect on tumors [1]. Curcumin possesses anti-tumor, anti-oxidant, and antiproliferative properties [2]. Curcumin may exert its effect by binding to various biomolecules involved in essential processes, including enzymes, transcription factors and signaling molecules [3]. Curcumin engages with several cell-signaling pathways and inhibits cancer cell proliferation. Recently, it has been seen that curcumin undergoes an autoxidative transformation at physiological pH leading to the generation of oxidative derivatives of curcumin [4] We have shown that the oxidation of curcumin is an essential step for its inhibitory effects against CagA, one of the major *H. pylori* virulence factors [5]. This study indicated that short-lived oxidative products of curcumin are active compounds against CagA.

*Helicobacter pylori* (*H. pylori*), a bacterium, causes gastric cancer and peptic ulcers in humans, and more than half of the human population is infected with *H. pylori.* It is a gram-negative, microaerophilic bacteria that colonizes the gastric epithelium [6]. Infection happens via the gastric-oral route in the host, and it is often acquired in childhood. Virulence factors secreted by *H. pylori*, such as cytotoxic associated gene (CagA) and vacuolating associated gene (VacA) are essential for the pathogenesis of the bacteria [7]. The CagA protein has a molecular weight of 120–140-kDa and is translocated in to the host cell via the type IV secretion system after bacterial adhesion [8]. It has been recently shown that oxidation of curcumin is required for inhibition of CagA translocation and phosphorylation [5].

VacA, another major virulence factor produced by *H. pylori* causes vacuolation of cultured cells and has been associated with gastric carcinogenesis [9]. *H. pylori* secretes VacA as a 140-kDa cytotoxin that matures as 88-kDa toxin 88-kDa monomer and further this monomer get fragmented into a 33-kDa N-terminal domain (p33) and a 55-kDa C-terminal domain (p55) by restricted proteolysis (p55) [10]. Experiments with intracellularly produced VacA toxin demonstrated that p33 domain and p55 domain are essential for cell vacuolation activity of VacA (residue 1–422) [11]. VacA sequence is polymorphic in three areas, an N-terminal signal zone (s1or s2), an intermediate zone (i1 or i2) at the C-terminus of p33, and a p55 mid zone (m1 or m2) [12]. Intramolecular interactions between the N-terminal sections of p33 in neighboring protomer, and contact between p33 of one protomer and an adjacent p55 arm of other protomers, are required for oligomerization in hexameric VacA [13]. This region includes amino acid residues 346 and 347 of the p55 domain, which is crucial in VacA oligomerization [14]. Polyphenol compound such as green tea polyphenols (GTPP), hop bract tannin (HBT) and catechin inhibit ion channel formation and VacA vacuolation activity [15].

In this study, in silico, and in vitro methods have been used to understand the effects of oxidized products of curcumin on VacA protein. Cyclobutyl cyclopentadione and dihydroxy cyclopentadione showed higher inhibitory action (binding affinities) for VacA cytotoxin than curcumin. In vitro study indicated that curcumin could prevent the vacuolation activity in AGS cells. In contrast, non-oxidizable derivatives of curcumin like acetyl curcumin and diacetyl curcumin could not be able to inhibit the vacuolation activity in AGS cells.

## 2. Results

### 2.1. The m1 and m2 Alleles of VacA Protein of H. pylori Are Similar at the 3D Structure Level

Amino acid sequences of the m1 and m2 alleles were aligned against reference sequences of VacA protein (Accession No. P55981). We observed that 84% of the amino acid sequence from m1 and 81% of the amino acid sequence of the m2 allele are similar to the reference sequence of VacA protein amino acid sequence [16] (Figure 1A). To see the structural similarity between the M1 and M2 allele of VacA protein at 3D structure, we created a homology structure using the I-TASER server tool. The superimposition of m1 and m2 VacA protein 3D structures showed a low RMSD value of 0.1 A for the whole structure, indicating a high level of overall structural similarity (Figure 1B). Further analysis of the structure revealed dissimilarities in the mid-region of m1 and m2 VacA protein. The region from amino acids 715–740 and 765–811 residues showed a higher value of RMSD (3.0 Å), indicating a higher degree of dissimilarities between m1 and m2 VacA proteins in the mid-region (Figure 1B).

### 2.2. Curcumin binds at Five Different Sites in the p55 Domain of the VacA Protein

To identify the potential binding sites in VacA protein, the 3D structure of VacA protein was prepared using I-TASSER. All-important regions of VacA including p33, p55, and the auto transporter region (Figure 2A,B) were included in the analysis. COACH servers were used to predict the VacA–curcumin binding sites. Based on C-score (0.99, range 0–1) COACH server used human GAPDH protein (6YNF) as a template for binding site prediction [17]. We observed the five-potential binding sites in VacA protein with high confidence scores. The interaction of curcumin with residues of VacA for each site is shown in Table 1. Surprisingly all the five predicted ligand-binding sites were present within the p55 domain of VacA protein (Figure 2C). We observed binding sites 1, 2, 3, and 4 consisting of amino acid residues 483–810 representing the mid-region of m1 and m2 VacA protein. Binding site 5, comprised of amino acid residues 395–435, was present in a region essential for the vacuolation activity of VacA protein (Figure 2C).

### 2.3. Oxidative Derivatives of Curcumin Show better Binding Affinity to VacA Compared to Curcumin

Oxidation of Curcumin is an essential step for its action against pathogenic activities of *H. pylori* infection. The molecular docking study of the p55 domain of VacA protein with curcumin and their oxidative derivatives such as dihydroxy cyclopentadione, hemiacetal cyclopentadione, ketohydroxy cyclopentadione, cyclobutyl cyclopentadione, and degradation products such as vinyl ether, diguaiacol, bicyclopentadione, was performed. We also performed docking studies with non-oxidizable curcumin such as acetyl curcumin and diacetyl curcumin. Docking analysis showed that binding affinities (*glide* score) of curcumin and oxidative derivatives of curcumin were observed between −0.1 to −7.349 kcal/mol at all predicted binding sites.

For all the predicted binding sites of p55, dihydroxy cyclopentadione and cyclobutyl cyclopentadione showed a higher binding affinity for VacA protein compared to curcumin at site1, 2, 3, and 5 (Figure 3).

The cyclobutyl cyclopentadione was found to interact most efficiently with p55 domain of VacA protein (−6.823,−7.349,−6.813,−5.926 and −5.325 kcal/mol) at active site 1, 2, 3, 4 and 5 respectively compared to curcumin (−3.805,−3.128,−3.206,−5.42 and −3.591 kcal/mol). Site 5 of p55 domain of VacA includes the region that is involved in the *H. pylori*-mediated vacuolation. We observed that site 5 possesses a higher binding affinity for cyclobutyl cyclopentadione (−5.325 kcal/mol) compared to curcumin (−3.591 kcal/mol) (Figure 3). As shown in Figure 4A,B, cyclobutyl cyclopentadione interacts with the VacA protein at the site 5 via polar interactions involving residues ASN422, SER433, SER434, GLN398, ASN397, and SER396. Cyclobutyl cyclopentadione formed H-bond with GLN424 and GLY427 with bond lengths of 1.85 and 2.63 Ǻ, respectively (Figure 4A,B). The non-polar hydrophobic interaction is also observed with TYR428, ALA429, LEU430, ALA435, GLY401, VAL421, LEU395, and LY427 residues (Figure 4B). The binding of cyclobutyl cyclopentadione indicates interaction with the beta-sheet and the loop critical for interaction with p33 and vacuolation activity of VacA protein (Figure 4C). Curcumin formed polar interaction with ASN397, SER396, GLN398, GLN424, and Ser433 residue of VacA protein (Figure 4D,E). Curcumin also formed H-bond with GLN 424 and ASN 440 instead of GLY427 with bond lengths of 2.45 and 2.14 Ǻ, respectively (Figure 4E). The non-polar hydrophobic interaction with curcumin is observed with, Val421, TYR428, ALA429, LEU430 and ALA405 (Figure 4E). Curcumin is involved in the binding interaction with the only near-to-loop region as shown through binding to different amino acid residues compared to cyclobutyl cyclopentadione (Figure 4F). The binding of curcumin and cyclobutyl cyclopentadione with p55 affects the binding of p55 and p33 and thus may also affect the vacuolation activity of the VacA protein.

### 2.4. In-silico Analysis Predicted Cyclobutyl Cyclopentadione Inhibits p33 and p55 VacA Subunits Interaction

Dimerization of p33 and p55 domains is an essential step in the oligomerization of VacA protein [18]. The binding site 5 includes the amino acid sequences where p33 and p55 domains interact to form the dimer and oligomer. Molecular docking of the p33 and p55 domains of VacA showed a higher HADDOCK score (Haddock score = −71.6 ± 5.8). Interacting interface analysis of the p33–p55 complex revealed that 23 amino acid residues of the p33 domain and 24 amino acid residues of the p55 domain are involved in the interaction of both the domains. The binding interaction revealed that the p33 and p55 domains interacted by forming one salt bridge, six H-bonds, and sixty non-polar bonds, as represented in Figure 5A. The binding of the cyclobutyl cyclopentadione with the p55 reduces the binding affinity between the p55 and p33 (HADDOCK score = −36.9 ± 2.8). This indicates a possible dimerization disruption between the p33 and p55 when cyclobutyl cyclopentadione is bound with p55 (Figure 5B). Further Interacting interface analysis of the p33 and p55-cyclobutyl cyclopentadione complex revealed interaction of only seven amino acids of the p33 domain with nine amino acids of the p55 domain (Figure 5B). The interface analysis revealed that the p33 and cyclobutyl cyclopentadione bound p55 domain interacted by forming one salt bridge, two H-bonds, and six non-polar bonds, as represented in Figure 5B. Whereas curcumin bound p55 showed higher binding affinity with p33 (HADDOCK score = −46.4 ± 3.7) compared to p55-cyclobutyl cyclopentadione complex (Figure 5C). Interface analysis of the p33 and p55-curcumin complex showed an interaction between eight amino acids of the p33 domain with twelve amino acids of curcumin bounded p55 domain (Figure 5C). The interaction revealed that the p33 and curcumin bounded p55 domain interacted with each other by forming one salt bridge, four H-bonds, and sixteen non-polar bonds, as represented in Figure 5C. Figure 5D shows disruption of p33–p55 interaction by cyclobutyl cyclopentadione. This suggests that cyclobutyl cyclopentadione may also inhibit the dimerization and thus oligomerization of VacA protein.

### 2.5. Oxidation of Curcumin Is Required to Inhibit VacA-mediated Vacuolation in AGS Cell Lines

To validate our in-silico results, we studied the level of *H. pylori* mediated vacuolation on gastric cancer cell line (AGS) pre-treated with curcumin and non-oxidizable curcumin derivatives, i.e., acetyl curcumin and diacetyl curcumin. Examination of *H. pylori*-infected gastric epithelial cells under the inverted microscope showed increased vacuole formation compared to uninfected cells (Figure 6A). We observed that both doses of curcumin caused a decrease in vacuole formation, while non-oxidizable curcumin acetyl and diacetyl curcumin failed to reduce the vacuole formation (Figure 6A). We further quantified the vacuoles by counting the cells and neutral red dye uptake assay. We observed a significant increase in vacuolation of *H. pylori*-infected gastric epithelial cells in an MOI-dependent manner (Figure 6B,C).

Both doses of curcumin showed a significant reduction of vacuolation and dye uptake in a dose-dependent manner. On the other hand, acetyl and diacetyl curcumin failed to reduce the *H. pylori*-induced vacuolation activity at 5 µM, and 20 µM concentrations. It showed similar vacuolation as observed in the *H. pylori*-infected untreated gastric cancer cell line (Figure 6B,C).

The failure of non-oxidizable curcumin’s derivatives to inhibit the VacA-mediated vacuolation and in silico analysis suggests that oxidation and thus the presence of oxidized curcumin, particularly, cyclobutyl cyclopentadione, is necessary for the inhibitory action of curcumin on VacA-mediated vacuolation.

## 3. Discussion

Curcumin, a natural polyphenolic compound, has an inhibitory effect on *H. pylori* growth. *H. pylori*, unlike other bacteria such as *E. coli*, fails to evade the inhibitory effects of curcumin due to its inability to reduce this compound [5]. This is attributed to the absence of the reductase enzyme CurA, suggesting that curcumin reduction may contribute to its inactivity [17]. While reduction of curcumin is associated with its reduced activity, oxidation of curcumin has been implicated in its enhanced biological activity. Recently we have shown that curcumin oxidation is essential for inhibiting *H. pylori* growth, phosphorylation, and translocation of one of its major virulence factors, CagA [5].

Curcumin’s pleiotropic actions are linked to its β-diketone moiety (nucleophilic addition), the dicarbonyl moiety (metal chelator), and the phenolic hydroxyl group (as an H-donor/anti-oxidant) [19,20]. Although whether these structural characteristics are enough to account for diverse cellular functions of curcumin is debatable. To explain the polypharmacology of curcumin and the disparity between its low plasma levels and it’s known in vivo effects, there is a notion that oxidative metabolites of curcumin mediate at least some of its biological effects [21,22]. A similar theory has been tied to the green tea catechins and their metabolites, ellagitannins, and urolithin metabolites. However, curcumin’s known conjugated, cleaved, or reduced metabolites are inactive or less active than curcumin, in contrast to the bioactivity of the ellagitannin and catechin metabolites [23].

Oxidative transformation of curcumin produces various metabolites such as dihydroxy cyclopentadione, hemiacetal cyclopentadione, ketohydroxy cyclopentadione, cyclobutyl cyclopentadione, spiro epoxide cyclopentadione, vinyl ether, guaiacol and final by-product dicyclopentadiene [4]. It is known that autoxidation of curcumin is one of the pathways that lead to curcumin metabolism. The biological functions of oxidative products of curcumin have not been fully understood. The α, β-unsaturated carbonyl of curcumin has been considered the only electrophile of the parent compound involved in interaction with the host proteins. However, curcumin’s oxidation products give additional electrophilic sites, implying that the parent compound’s enone is not the only functional electrophile. The oxidative intermediate cyclopentadione compounds of curcumin are less stable and interact with proteins. It has been shown that cyclobutyl cyclopentadione has better thiol reactivity than curcumin. Interaction with proteins can stabiles the oxidative unstable cyclopentadiene compounds by forming an adduct with target proteins. A future study is needed to ascertain the presence of cyclopentadiene adduct in the proteins isolated from curcumin-treated biological samples.

In the present study, in silico analysis showed that intermediate cyclopentadione compounds have better binding affinity to VacA at all sites except site-4 compared to curcumin. This differential binding affinity of intermediate cyclopentadione compounds could be attributed to the differences in the structure of intermediate compounds and, thus, differential reactivity of the cyclopentadione compounds. In silico analysis indicated that cyclobutyl cyclopentadione shows maximum binding affinity with VacA compared to other cyclopentadione. The increase in the reactivity of cyclopentadione compounds could be attributed to the chemical structure of the transformed compounds. Oxidative metabolism of curcumin causes cyclization of a carbon-carbon bond connecting C-1 and C-7 of the heptadienedione chain of curcumin and increases the reactivity of cyclopentadione compounds to proteins.

Further oxidative transformation of ketohydroxy cyclopentadione leads to the formation of a four-membered carbon ring and a cyclopentadione in cyclobutyl cyclopentadione. Biotransformation of curcumin contributes to the more electrophilic and nucleophilic sites in cyclobutyl cyclopentadione and spiro epoxide, which may be responsible for its enhanced reactivity with the VacA than the curcumin. The inability of non-oxidizable curcumin to generate cyclopentadione ring and/or four-member carbon ring containing intermediate compounds reflects its inability to inhibit the vacuolation activity of VacA.

Dimerization of the p33 and p55 domains of VacA protein is required for the vacuolation activity of VacA [24]. It has been reported that amino acid residues 346 and 347 of VacA protein are essential for the oligomerization of VacA [14]. Dimerization is a crucial step for the oligomerization of p33 and p55 monomers. It has been shown that p55 with deletion of amino acid residues 346 and 347 did not form the p33–p55 complex in the present study, in silico analysis has shown that the binding affinity of p33 and p55 (HADDOCK score = −71.6 ± 5.8), domains can be disrupted by binding of p55 with curcumin (HADDOCK score = −46.4 ± 3.7) and cyclobutyl cyclopentadione (HADDOCK score = −36.9 ± 2.8). The binding site-5 of VacA contains the amino acid residues responsible for the dimerization of p33 and p55. Although curcumin and cyclobutyl cyclopentadione does not bind to residue 346 and 347 of the p55 domain, as shown in this study, their binding to the vicinity may provide a steric hindrance for the interaction of p33 with p55. In vitro study indicated that curcumin could prevent the vacuolation activity in AGS cells. In contrast, acetyl curcumin and diacetyl curcumin could not inhibit the vacuolation activity in AGS cells. Synthesis of these oxidized curcumin derivatives might potentially provide new therapeutic drug molecules for inhibiting *H. pylori*-mediated pathogenesis.

Oxidative derivatives of curcumin are short-lived and unstable compounds under physiological conditions [25]. The cyclopentadione derivatives showed a better binding affinity for VacA than curcumin itself. We have shown that non-oxidizable curcumin derivatives such as acetyl curcumin and diacetyl curcumin could not inhibit VacA-induced vacuolation. However, synthesis or invitro stabilization of cyclobutyl cyclopentadione is required to understand it’s biological effects on the Vac A or any other biological events. This could be possible by chemically modifying the compound to increase their stability under physiological conditions. Synthesis of these compounds could provide a new class of drugs against *H. pylori* associated pathology.

## 4. Method

### 4.1. Sequence Alignment of m1 and m2 Isoforms of VacA Protein

The amino acid sequences of m1 and m2 isoforms were obtained from the NCBI database (accession no. Q482456~m1), accession no. Q48253~m2) (https://www.ncbi.nlm.nih.gov/ (accessed on 12 November 2021). MUSCLE (Multiple Sequence Comparison by Log-Expectation) software [26,27] was used to align the protein sequences, which were then analyzed in Bioedit (http://www.mbio.ncsu.edu/BioEdit/bioedit.html (accessed on 12 November 2021) [27].

### 4.2. Structure Retrieval and Binding Site Prediction

The 3D X-crystallographic structure of the crystal structure of VacA p55 protein (PDB ID: 2QV3) was retrieved from RCSB Protein Data Bank (https://www.rcsb.org (accessed on 15 March 2022)) and was used for molecular docking. COACH meta-server was used to predict the potential protein-ligand binding sites of VacA protein [28].

### 4.3. Protein and Ligand Preparation

The 2D chemical structure of curcumin was retrieved from the PubChem database. However, 2D chemical structures of dihydroxy cyclopentadione, hemiacetal cyclopentadione, ketohydroxy cyclopentadione, cyclobutyl cyclopentadione, spiro epoxide cyclopentadione, vinyl ether, diguaiacol, bicyclopentadione, acetylcurcumin, and diacetylcurcumin compounds were prepared using ChemSketch™ version 2019 (ACD/Labs’ software, Toronto, Canada) [21]. The 2D chemical structures of compounds were converted into 3D chemical structures using Schrodinger 2018-2 [29]. Ligands were prepared using the big prep module in the Schrodinger release 2018-2 (Schrödinger, LLC, New York, NY, USA, 2018) [29,30] At physiological pH of 7.0 ± 0.2, the various ionization states for each ligand structure were generated, and low-energy 3D structures with proper chiralities were generated. The ligands were minimized using an OPLS3 (optimum potentials for liquid simulations) force field, and all other parameters were left at default [31,32]. VacA protein was prepared using Maestro’s Schrodinger protein preparation wizard [33]. The glide module of the Schrodinger was used to add hydrogen atoms, remove water molecules beyond 5 Å from the binding site, and for the receptor grid generation [28,32]. A receptor grid of 10 Å × 10 Å × 10 Å was created on identified binding site residues of the VacA protein. Protein was optimized at neutral pH for all atoms using the OPLS3 force field.

### 4.4. VacA Protein-ligand Molecular Docking

A glide ligand-docking procedure was followed to evaluate binding interactions between the VacA p55 protein and the ligands (curcumin, its oxidative derivatives, and non-oxidizable derivatives). The prepared ligands were selected and docked with the predicted potential binding sites of p55 VacA protein. Each ligand was docked into every refined low-energy conformation of the p55 domain of the VacA protein using the rigid receptor docking method [34]. The Docking score was used to evaluate the binding affinity of the protein-ligand docked complexes. The best-docked position with the lowest docking score value was recorded. Finally, the most favorable ligand orientation with the lowest free energy (highest binding affinity) was selected for further structural analysis of the protein-ligand complex using the glide module of the Schrödinger suite [34].

### 4.5. Docking Study of p33 and p55 Domain

HADDOCK web server was employed for protein-protein docking analysis [35]. The model with the lowest HADDOCK score was selected for further analysis of intermolecular interactions of the p33 and p55 domains of the VacA protein. Pymol was used to assess and illustrate structural models with the lowest energy score. PDBsum (http://www.ebi.ac.uk/pdbsum (accessed on 9 March 2022)) was used to show intermolecular interacting residues across the protein complexes and was represented as a pictorial interaction map (http://www.pymol.org/pymol (accessed on 10 March 2022)). Further, PDBsum was used to demonstrate intermolecular bonded and non-bonded interactions for H-bonds, salt bridges, and hydrophobic interactions. The RCSB PDB (https://www.rcsb.org/ (accessed on 15 March 2022) is used to estimate the membrane of the VacA protein’s hexamer structure.

### 4.6. H. pylori Growth Condition

*H.pylori* strain 26,695 cultured on BHI (brain heart infusion) agar (with 10% sheep blood, 2% Isovitalex (BD bioscience, Singapore), 4 µg/mL Amphotericin B (Sigma, St. Louis, MO, USA), 4 µg/mL Trimethoprim (Sigma, St. Louis, MO, USA), 4 µg/mL Vancomycin (Sigma, St. Louis, MO, USA) for 3–4 days at 37 °C with 10% CO_2_ [35]_._
*H. pylori* (strain 26,695) was cultured in Brucella broth (Himedia, PA, USA) supplemented with 10% fetal bovine serum (Himedia, PA, USA) for 16 to 18 h at 37 °C with 5% CO_2._

### 4.7. H. pylori Co-culture with AGS Cells

AGS (human gastric epithelial cells) cells (ATCC), isolated from gastric cancer patients, were cultured in F-12 Ham’s (Himedia) supplemented with 10% fetal bovine serum (Himedia), L-glutamine (2 mM BD Biosciences) and 1% penicillin-streptomycin (Hi-media) at 37 °C with 5% CO_2._
*H. pylori* strain 26,695 co-cultured with AGS cells at a multiplicity of infection (MOI) 10:1 and 100:1 for 24 h at 37 °C with 5% CO_2_ [36]. For some experiments, pre-treatment with two different doses (5 mM and 20 mM) of curcumin, acetyl curcumin, and diacetyl curcumin with AGS cells was performed.

### 4.8. AGS cells Infection with H. pylori and Neutral Red Dye Assay

AGS cells were co-cultured with *H. pylori* supernatants for 24 h at 37 °C with 5% CO_2_. AGS cells stained with 0.3% BSA (SRL) contain 0.05% neutral red dye (sigma) to analyze the vacuolation activity of *H.pylori* infected AGS cells [36,37]. The neutral red dye was extracted with 70% ethanol and 0.3% HCl. The absorbance in the spectra Count Microplate Photometer was read at 540 nm wavelength after subtracting the OD from uninfected control cells [38].

### 4.9. Statistics

The data were expressed as a mean ± standard deviation. Two-way ANOVA (Analysis of variance) tests with Tukey’s multiple comparison test were used for all statistical tests. A statistically significant correlation was evaluated as a *p*-value of <0.001.

## 5. Conclusions

Oxidative derivatives of curcumin are short-lived compounds and are unstable at physiological pH. During in-silico studies, these derivatives showed a better binding affinity for VacA toxin particularly, cyclobutyl cyclopentadione, which showed the highest binding affinity at most of the binding sites. Moreover, cyclobutyl cylopentadione binds to a region of VacA, which has been proved critical for the toxin’s vacuolation activity. Apart from this, cyclobutyl cyclopentadione also inhibits the interaction of the p33 domain of VacA with its p55 domain, preventing their dimerization and thus preventing oligomerization of toxins. Our in vitro results indicate that oxidation of curcumin is essential for its activity against VacA-mediated vacuolation as non-oxidizable curcumin derivatives failed to inhibit vacuolation. The in silico and in vitro data indicates that oxidative products may be responsible in part for inhibitory effect of curcumin on Vac A.

## Figures and Tables

**Figure 1 molecules-27-06727-f001:**
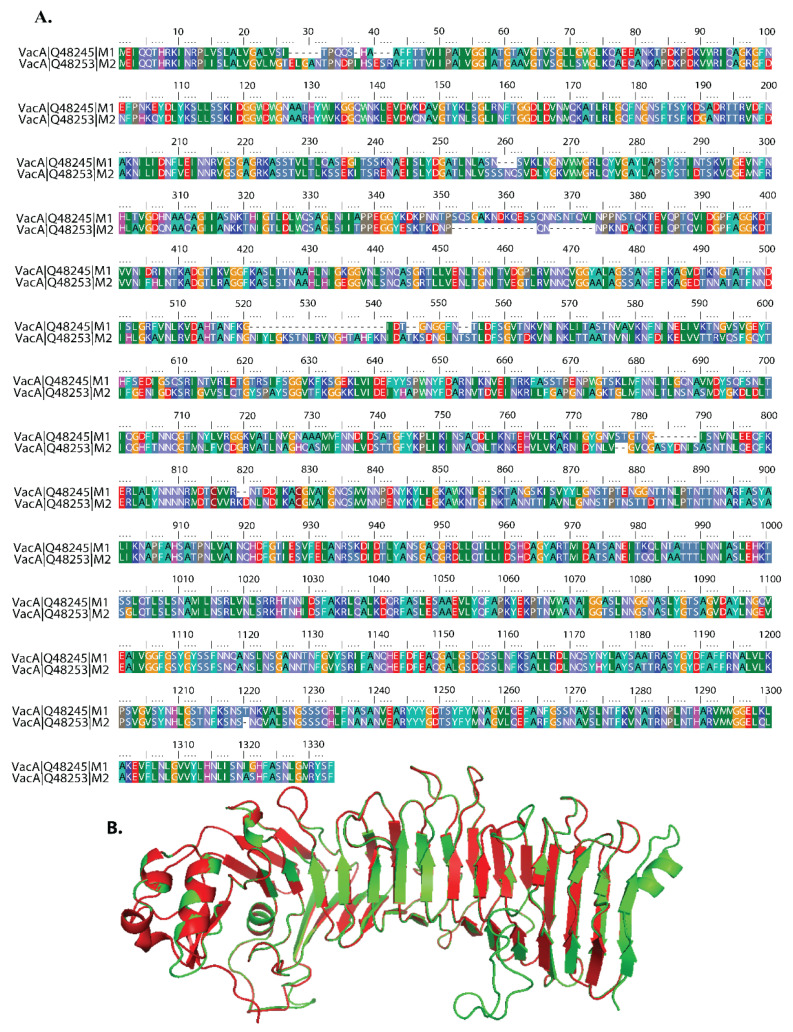
Sequence and structure of VacA isoform. (**A**) Comparison of amino acid sequences of M1 and M2 isoforms of VacA protein of *Helicobacter pylori*. Corresponding accession nos. Q48245 (*H. pylori* strain 60190, m1 VacA) and Q48253 (strain Tx30a, m2 VacA). Amino acid color code generated by ClustalW within BioEdit software was used. Consensus residues ("Clustal cons" line) were generated by ClustalW. The numbers at the beginning and end of each line are for reference only and do not correspond to the original numbers of individual amino acid sequences recorded in published reports. Hyphens indicate gaps in the alignment at that position. (**B**) The ribbon image shows the superimposition of the I-TASSER modelled m1 protein (green color) with m2 protein (red color).

**Figure 2 molecules-27-06727-f002:**
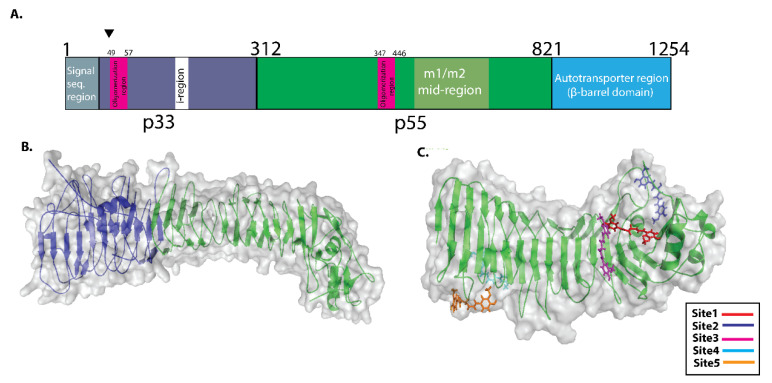
Schematic depiction of VacA protein structure and potential binding sites for Curcumin. (**A**) p33 domain of VacA protein (1–311 amino acid sequence region) (shown in blue colour). The p55 domain of VacA (312–821 amino acid sequence region) (shown in green colour). VacA p33 region includes an initial signal sequence, an oligomerization region in the middle, and an intermediate (“i”) region at the end. The mid-region amino acid sequences of the p55 domain of VacA protein define m1 and m2 isoforms (**B**) The VacA p33 and p55 are three-dimensional structures. (The p33 and p55 domains are shown in blue and green colors, respectively. (**C**) COACH server predicted potential binding site present in p55 domain of VacA protein. Site1, site 2, site 3, site 4, and site 5 are shown in different colors.

**Figure 3 molecules-27-06727-f003:**
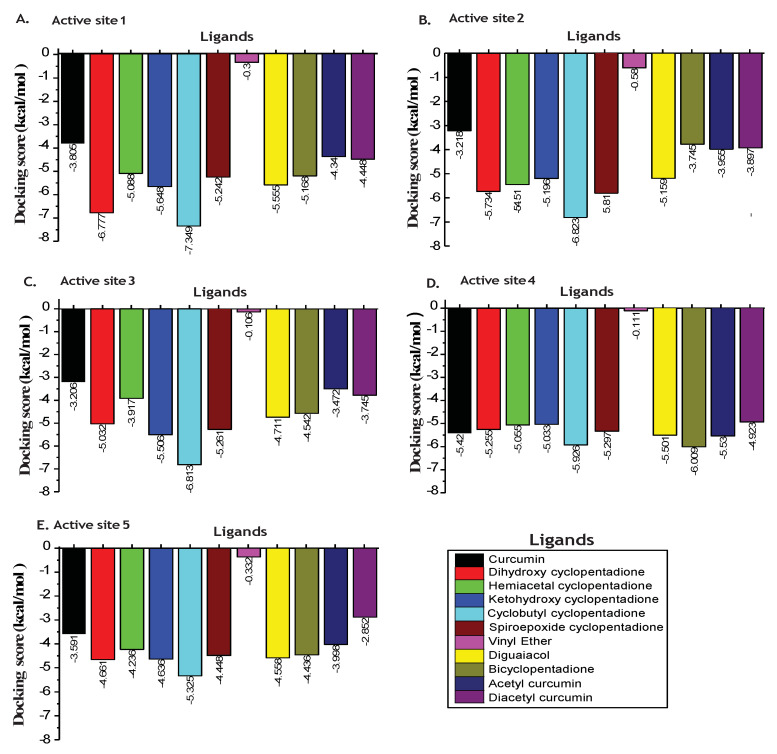
Binding affinity of curcumin and it’s oxidative products to p55 domain of VacA protein. Histogram graphs showing docking scores of curcumin, its oxidative derivatives, and non-oxidizable derivatives with the different binding sites. Different colors represent different ligands. (**A**) Binding affinity of curcumin, its oxidative derivatives, and non-oxidizable derivatives with site 1. (**B**) Binding affinity of curcumin, its oxidative derivatives, and non-oxidizable derivatives with site 2. (**C**) Binding affinity of curcumin, its oxidative derivatives, and non-oxidizable derivatives with site 3. (**D**) Binding affinity of curcumin, its oxidative derivatives, and non-oxidizable derivatives with site 4. (**E**) Binding affinity of curcumin, its oxidative derivatives, and non-oxidizable derivatives with site 5.

**Figure 4 molecules-27-06727-f004:**
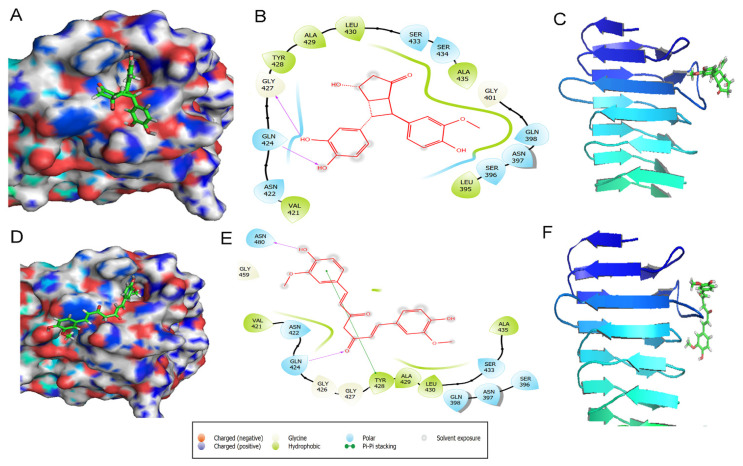
p55 and ligand Cyclobutyl cyclopentadine and Curcumin interaction. Protein-ligand interaction diagrams for Cyclobutyl cyclopentadiene: 3D crystallized structure, secondary structure, and 2D interaction diagrams (**A**) VacA binding site 5 complexed with Cyclobutyl cyclopentadiene. (**B**) In the 2D protein-ligand interaction diagram different color show types of bond interaction i.e., polar, hydrophobic, pi − pi interactions, etc. (**C**) Interaction of cyclobutyl cyclopentadione. with secondary structure of p55 domain of VacA. Protein-ligand interaction diagrams for curcumin: 3D crystallized structure, secondary structure, and 2D interaction diagrams (**D**) VacA binding site 5 complexed with curcumin. (**E**) In the 2D protein-ligand interaction diagram different color show types of bond interaction i.e., polar, hydrophobic, pi-pi interactions, etc. (**F**) Interaction of curcumin with secondary structure of p55 domain of VacA.

**Figure 5 molecules-27-06727-f005:**
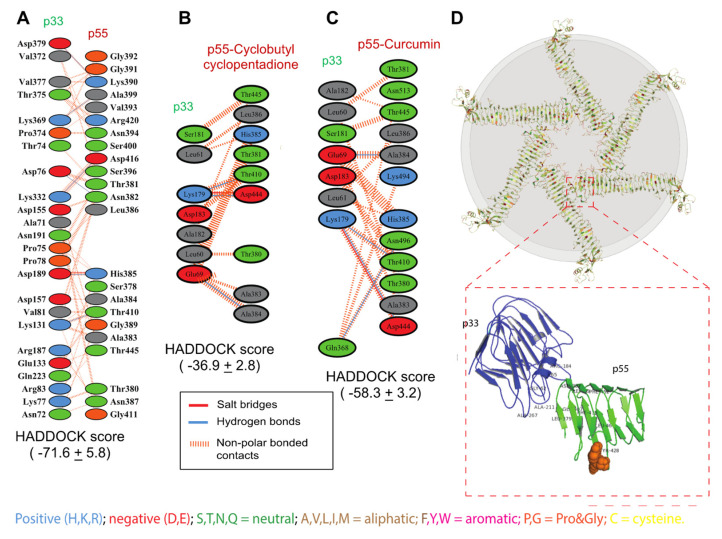
Effect of Cyclobutyl cyclopentadine and Curcumin on p33 and p55 docking interaction and vacA protein hexamer. (**A**) Schematic representation of binding interaction between the p33 and p55 domains of VacA protein. Amino acid residues involved in the interaction between p33 and p55 domains are shown in different colors based on their chemical structure. Different bonding interactions between amino acids are shown in different colors i.e., salt bridges, hydrogen bonds, etc. (**B**) Schematic representation of binding interaction between the p33 and p55-Cyclobutyl cyclopentadione complex. Amino acid residues involved in the interaction between p33 and p55-Cyclobutyl cyclopentadione complex are shown in different colors based on their chemical structure. Different bonding interactions between amino acids are shown in different colors i.e., salt bridges, hydrogen bonds, etc. (**C**) Schematic representation of binding interaction between the p33 and p55-curcumin complex. Amino acid residues involved in the interaction between p33 and p55-curcumin complex are shown in different colors based on their chemical structure. Different bonding interactions between amino acids are shown in different colors i.e., salt bridges, hydrogen bonds, etc. (**D**) A blueprint of VacA protein hexamer formed by VacA protomers. p33 (blue colored) and p55 (green colored) domains involved in hexamer formation.

**Figure 6 molecules-27-06727-f006:**
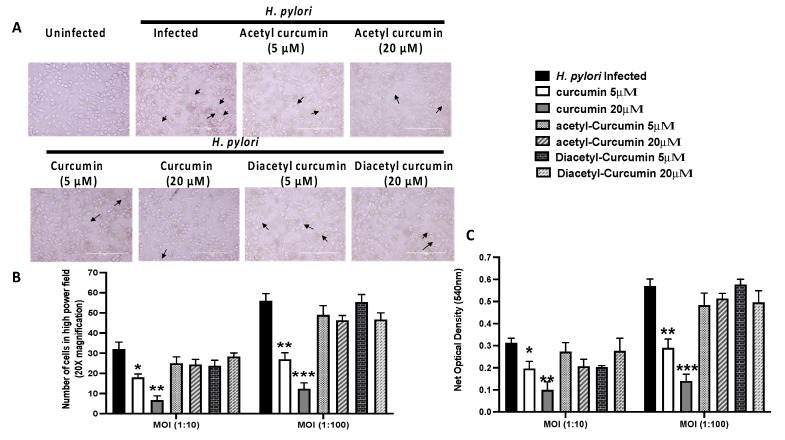
Effect of curcumin and non-oxidizable derivatives on *H. pylori* mediated VacA vacuolation activity in AGS cells (* *p* value < 0.05; ** *p* value < 0.01; *** *p* value < 0.001 vs *H. pylori* infected). Data are expressed as means ± SEM, *n* = 3 independent replicates. (**A**) Light microscopy images showed uninfected AGS cells. The black arrow indicated the vacuolation region in *H.pylori*-infected AGS cells pretreated with curcumin and non-oxidizable curcumin. (**B**) The histogram graph represented the inhibition of vacuolation activity in *H. pylori-infected* AGS cells at two MOI (10:1, 100:1) and pretreated with curcumin and non-oxidizable curcumin. (**C**) The histogram graph showed neutral red uptake by AGS cells at OD 540 nm in *H. pylori-infected* AGS cells at two MOI (10:1, 100:1) and pretreated with curcumin and non-oxidizable curcumin. (* *p* value < 0.05; ** *p* value < 0.01; *** *p* value < 0.001 vs. *H. pylori* infected). Data are expressed as means ±  SEM, *n* = 3 independent replicates.

**Table 1 molecules-27-06727-t001:** Curcumin binding sites in p55 domain of VacA.

Site	Score	Predicted Binding Site Residues
Site 1	0.97	Asn618, Val620, Arg641, Val643, His669, Thr672, Phe674, Gly675, Ile676, Pro677, Lys680, Tyr729, Asn732, Asn733, Arg734, 737, Cys738, Val739, Arg741, Asp745, Ala748, Cys749, Vla752, Ala483, Asn529, Phe545, Asn546, Arg574, Phe565, Lys619
Site 2	0.951	Asn666, 668, 674, Thr675, Gly704, Asn705, Thr708, Gly709, Thr710, Asn711, Gly712, Ile713, Ser714, Val716, Asn717, Leu718, Glu720, Gln721, Lys723, Glu724, Arg725, Asp736, Asn737, Lys772, Trp774, Asn775, Ile776, 778, 779, 780, 803, Thr805, 806, 807, Pro808, Thr809, 810
Site 3	0.748	Tyr574, Ala595, Ser596, Pro599, Glu600, Pro602, Asp622, 623, Ser624, Asn643, Tyr644, Leu645, Arg647, Lys680, Ile681, Asn682
Site 4	0.642	Phe483, Asn484, Thr485, Ser504, Thr505, Asn506, Glu528, Thr529, His531
Site 5	0.541	Leu395, Ser396, Asn397, Gln398, Gly401, Arg402, Val421, Asn422, Thr428, Ala429, Leu430, Ser433, Ser434, Ala435

## Data Availability

Not applicable.

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
