# Peer review of "Oxidative Products of Curcumin Rather Than Curcumin Bind to Helicobacter Pylori Virulence Factor VacA and Are Required to Inhibit Its Vacuolation Activity"

_molecules, 2022, doi:10.3390/molecules27196727_

Round 1

Reviewer 1 Report

The article by Maya et al, describing "oxidative products of curcumin and its binding to H.pylori virulence factor". The article contains a major portion of computational data of curcumin. The article overall is with the average score but the article is full of mistakes and blunders from start till end. A real major revision can only give a hope to this paper.

1.      The title should not contain abbreviation “H.pylori”.

2.      In-vitro results are not well discussed in abstract. It should be discussed.

3.      The references in the text and bibliography is not set according to the journal style. Kindly look at the instruction for authors and prepare the article according to the journal style.

4.      There are so many old references in the article. The authors should use the updated references to cite in their paper.

5.      Fig.1 is fuzzy and all the words are not able to be read, the authors should provide the high resolution of an image.

6.      Fig.4 is totally wrong. The authors should provide the original and absolute figures. In this figure 4a and 4c are missing.

7.       The article is also very poorly written some sentences are very difficult to read, and there are many typo mistakes. Page.7: Second last sentence: (Both Doses)... The "doses" should be in small letters.

8. Conclusion is missing.

Author Response

1. The title should not contain the abbreviation “ pylori”.

We would like to thank the reviewer for pointing out, we have changed the H.  pylori abbreviation to Helicobacter pylori.

2. In-vitro results are not well discussed in the abstract. It should be discussed.

We appreciate the reviewer’s comment, now sentences have been included in the abstract section describing in vitro experiments.

3. The references in the text and bibliography are not set according to the journal style. Kindly look at the instruction for authors and prepare the article according to the journal style.

We have corrected and added the bibliography according to the journal style.

4. There are so many old references in the article. The authors should use the updated references to cite in their paper.

We highly appreciate the positive feedback and agree with the reviewer's comment. We have replaced old references with new ones, cited their paper, and updated them.

5. Fig 1 is fuzzy and all the words are not able to be read, the authors should provide the high resolution of an image.

We have updated Fig.1 with a high-resolution image. Each word is more readable now.

6. Fig 4 is totally wrong. The authors should provide the original and absolute figures. In this figure, 4a and 4c are missing.

We would like to thank the reviewer for pointing out the problem in Fig 4. We have named the interaction interface figure as Fig 4 B and E for the interaction of cyclobutyl cyclopentadione and curcumin with p55 respectively. We really regret the inconvenience. Now we have added a high-resolution figure in the text and also as a pdf separately.

7. The article is also very poorly written some sentences are very difficult to read, and there are many typo mistakes. Page.7: Second last sentence: (Both Doses)... The "doses" should be in small letters.

We appreciate the reviewer’s comment. We have corrected the sentence mistakes and Page.7: Both Doses of capital letters are replaced with small letters in both doses. Sentences have been modified for better readability.

8. Conclusion is missing.

We apologize for missing the conclusion. Now conclusion has been added in the manuscript.

Reviewer 2 Report

1.     Are there any other natural product derivatives that have been reported that inhibit the vacuolation activity of H. pylori.? Please discuss this in the introduction part.

2.     CagA and VacA are the important virulence factors secreted by H. pylori. Research has reported that oxidation of curcumin is required for inhibition of CagA, in this paper, the author showed that oxidized products of curcumin bind with VacA. Do the oxidized products of curcumin inhibit the activity of VacA?

3.    The quality of figure 4 is needed to improve. low resolution

4.  In figure 6, the concentration used in the figure and figure ligand is different, whether mM or μM of curcumin and non-oxidizable derivatives are used?

5.     Please add the p-value in figure 6B.

6.  The author could compare the binding affinity of curcumin and non-oxidizable derivatives on VacA in cellulo by CETSA assay.

7.     The author should discuss the novelty of this study based on the results.

8.     Are there any other virulence factors secreted by H. pylori?

9.     There are also some typos and grammatical errors.

Author Response

1. Are there any other natural product derivatives that have been reported that inhibit the vacuolation activity of pylori.? Please discuss this in the introduction part.

Polyphenol compound such as green tea polyphenols (GTPP), hop bract tannin (HBT), and catechin has been shown to inhibit ion channel formation and VacA vacuolation activity in gastric epithelial cells. A sentence has been discussed and is included in the introduction section with reference.

2. CagA and VacA are the important virulence factors secreted by H. pylori. Research has reported that oxidation of curcumin is required for inhibition of CagA, in this paper, the author showed that oxidized products of curcumin bind with VacA. Do the oxidized products of curcumin inhibit the activity of VacA?

We appreciate the question raised by the reviewer. The oxidized products of curcumin are not stable and hard to isolate or synthesize. The unstable nature of oxidized curcumin may be the reason for higher binding affinity to the p55 domain of the VacA protein.  Efforts are underway in the laboratory for the synthesis and stabilization of curcumin oxidized products. But in the present study, we have shown that acetyl and diacetyl curcumin that cannot be oxidized failed to inhibit the VacA-mediated vacuolation activities. We also showed that the binding affinity of nonoxidizable curcumins is much lower than oxidative products of curcumin.       

3. The quality of figure 4 is needed to improve. low resolution.

We appreciate the reviewer’s comment. We have replaced low resolution with high-resolution Figure 4.

4. In figure 6, the concentration used in the figure and figure ligand is different, whether mM or μM of curcumin and non-oxidizable derivatives are used?

We really apologize for the oversight. We have corrected the concentrations from mM to μM in both Figure 6 and Figure 6 legend for curcumin and non-oxidizable derivatives. 

5. Please add the p-value in figure 6B.

We highly appreciate the reviewer’s comment. We have added p-values for Figure.6

6. The author could compare the binding affinity of curcumin and non-oxidizable derivatives on VacA in cellulo by CETSA assay.

We agree with the reviewer’s comment about studying affinity using biophysical methods. In the present study, we have shown that oxidative products are binding to the p55 domain of Vac A protein rather than curcumin itself. In the in vitro culture, curcumin gets oxidized rapidly to cyclobutyl cyclopentadiene. In a cell-free culture system probably curcumin may not get oxidized and we would not be able to see the binding to the p55 domain. Presently, A study is underway in the laboratory for the synthesis and stabilization of cyclobutyl cyclopentadione.   

7. The author should discuss the novelty of this study based on the results.

We thank the reviewer for pointing out the novelty of the work. We have discussed the novelty of our work in the discussion section. This study highlights the fact that curcumin is not a biologically active molecule but the intermediate oxidative products are the molecules causing the biological effects. We have published previously that the oxidation of curcumin is essential for its interaction with another H. pylori virulence factor CagA (Front. Cell. Infect. Microbiol., 24 December 2021). Synthesis of these oxidized curcumin derivatives could potentially provide new therapeutic drug molecules for inhibiting H. pylori-mediated pathogenesis.

8. Are there any other virulence factors secreted by H. pylori?

CagA and Vac A have been shown to be associated with gastric cancer and are a major focus in H. pylori pathogenesis. There are three categories of H. pylori virulence factors 1. Colonization (Urease - for Neutralize gastric acid, Flagella Chemotaxis system- Bacterial movement to epithelial surface & deep gland, Adhesins (BabA, SabA, Lewis antigens, and OipA) - Adherence to gastric epithelial cells). 2. Immune escape (LPS & Flagella - Low immunogenicity Molecular mimicry Induce anti-inflammatory response, CagA & T4SS- Suppress phagocytosis Decrease antimicrobial peptide Induce tolerogenic dendritic cell Block effector T cell response, VacA- Suppress phagocytosis Induce tolerogenic dendritic cell Block effector T cell response and Arginase- Suppress ROS & NO Block effector T cell response). 3. Disease induction (CagA & T4SS, VacA, BabA, HtrA, DupA, IceA, and OipA).

9. There are also some typos and grammatical errors.

We would like to thank you for the reviewer's comment. We have corrected the grammatical errors and updated them.

Reviewer 3 Report

This study aims to investigate the effects of curcumin and its metabolites on Vac protein by using in silico and in vitro methods. In silico results show that oxidized products of curcumin have better binding affinity to Vac protein. The main idea of this research is interesting, however, verification in vitro or in vivo study is lack. In vitro study was only performed to test curcumin and its non-oxidized products. The non-oxidized products were not able to inhibit the vacuolation activity in AGS cells; and this doesn’t mean that oxidized products have better inhibitory ability. Therefore, I don’t think the conclusion of the manuscript is convincing. In addition, some spell mistakes should be checked throughout the manuscript.

Author Response

This study aims to investigate the effects of curcumin and its metabolites on Vac protein by using in silico and in vitro methods. In silico results show that oxidized products of curcumin have better binding affinity to Vac protein. The main idea of this research is interesting, however, verification in vitro or in vivo study is lacking. In vitro study was only performed to test curcumin and its non-oxidized products. The non-oxidized products were unable to inhibit the vacuolation activity in AGS cells; this doesn’t mean that oxidized products have the better inhibitory ability. Therefore, I don’t think the conclusion of the manuscript is convincing. In addition, some spelling mistakes should be checked throughout the manuscript.

We agree that the study lacks direct evidence that oxidized curcumins are inhibiting H. pylori-induced vacuolation in the gastric epithelial cells. Although we do provide in-vitro evidence that oxidation of curcumin is an essential step in the anti-vacuolation activity of the curcumin. We have modified our conclusion and have included this point along with our Insilco observations.  

Round 2

Reviewer 1 Report

The authors addressed all the points very clearly, i recommend for its publication in its current state.

Reviewer 3 Report

The authors discussed the unstability of oxidative derivatives of curcumin, which answered my previous question. It is fine in the present form.

In addition, please correct "invitro" into "in vitro".